# Observations on the Surface Structure of *Aurelia solida* (Scyphozoa) Polyps and Medusae

Valentina Turk [1],*, Ana Fortič [1], Maja Kos Kramar [1], Magda Tušek Žnidarič [2], Jasna Štrus [3], Rok Kostanjšek [3] and Alenka Malej [1]

[1] National Institute of Biology, Marine Biology Station Piran, 6330 Piran, Slovenia; Ana.Fortic@nib.si (A.F.); maja.kos.kramar@gmail.com (M.K.K.); Alenka.Malej@nib.si (A.M.)

[2] National Institute of Biology, Department of Biotechnology and Systems Biology, 1000 Ljubljana, Slovenia; Magda.Tusek.Znidaric@nib.si

[3] Biotechnical Faculty, University of Ljubljana, 1000 Ljubljana, Slovenia; jasna.strus@bf.uni-lj.si (J.Š.); Rok.Kostanjsek@bf.uni-lj.si (R.K.)

\* Correspondence: valentina.turk@nib.si

**Abstract:** The surface structures and mucus layers that form an interface between the epithelial layer of organisms and their external environment were studied in the bloom-forming moon jellyfish (*Aurelia solida*, Scyphozoa) from the northern Adriatic. The surface of the polyps revealed epithelial ciliated cells and numerous nematocysts, both non-discharged and discharged. Cilia were also the most prominent features on the surface of adult medusa, protruding from the epidermal cells and with microvilli surrounding the base. Histochemical methods and various microscopy techniques (light/epifluorescence and electron microscopy) confirmed the presence of abundant mucus around polyps and on the surfaces of adult medusa, and that the mucus contained acidic and neutral mucins. The observed mucus secretions on the exumbrella surface of the medusae were in the form of granules, flocs, and sheets. Scanning electron microscopy and transmission electron microscopy analyses confirmed the presence of various microbes in the mucus samples, but not on the epithelial surfaces of the polyps or the exumbrella of the medusae.

**Keywords:** *Aurelia solida*; polyp; medusa; surface microlayer; mucus; scanning electron microscopy (SEM); transmission electron microscopy (TEM); histology





## 1. Introduction

One of the successful strategies of planktonic organisms is to have a comparatively large, low-energy (gelatinous) body [1]. Such organisms usually have a transparent body that protects them from predators [2]. Furthermore, the surface structure is an important feature of these organisms in relation to their life strategy. In addition to the transparent body, specific surface structures with anti-reflective properties can achieve reduced visibility in the water column [3,4]. Such a structure can help suppress biofouling [5] and reduce debris and bubble adhesion [6]. Free-drifting scyphomedusae are among the characteristic gelatinous organisms.

Like most scyphozoans, members of the genus *Aurelia* are characterized by a bipartite life cycle with a perennial sessile polyp and free-swimming medusa [7]. These simple diploblastic organisms are therefore a suitable model for comparing surface structures in relation to different habitats and the different life strategies. Our model species was *Aurelia solida* (Browne, 1905), originally described as being from the Maldives Islands, in the Indian Ocean [8]. In addition to the Indian Ocean, *A. solida* is currently distributed in the Central Atlantic Ocean and Mediterranean Sea [7], as well as in the Adriatic Sea, where it forms widespread seasonal blooms [9]. It thus represents an easily accessible model organism that is also of interest due to its success as an introduced organism [10].

Ultrastructural studies of scyphozoans, especially of surface structures, are rare and usually part of broader cnidarian studies. Most have focused on mesoglea, gastric cavity, muscular, and sensory systems in various Medusozoa. Polyps have been studied by Calder [11], Hetschel and Hiindgen [12], Lauth et al. [13], while Branden et al. studied ephyrae [14]. The ultrastructure of some compartments in adult medusa has been investigated in comparative studies of the function, diversity, and evolution of the mesoderm or mesoglea [15–17], the skeleton [16], the anatomy of the muscular system together with elasticity [18,19], and the nervous system or rhopalia [20,21]. Chapman studied mesoglea and tentacles of the polyps of *A. aurita*, as well as the microanatomy of the umbrella margin [22–24]. These most basic studies provided certain information about the cellular organization of scyphozoans, but they are scattered and a detailed review is necessary. Recently, a more complex comparison of the nanostructure of the surface of the exumbrella of five pelagic cnidarians was presented by Hirose and co-workers [4].

In scyphozoans, as in many aquatic organisms, the body surface is covered with a mucus layer that forms a multifunctional hydrogel interface between the epithelial cells and the external environment. The production and secretion of mucus is an important mechanism for defense [25], nutrition, and other processes important for the health of the organism (desiccation, gas exchange, etc.) [26,27], including the immune system of animals [28]. Different species release different amounts of mucus, which has a variable composition of proteins, polysaccharides, lipids, and amino acids [29]. Mucus may contain more than 1200 proteins in extracellular or membrane-associated regions and variety of metabolites [30] that may act as a modulator of innate immunity against physical damage, threats from various contaminants, and microbial invasion [30–32].

Diverse microbial communities colonize the surface of marine organisms and form various types of associations with the host [33]. Interaction with marine invertebrates has been studied in a number of benthic organisms, particularly corals [34], sponges [35], but less so in other cnidarians. Recent studies of the medusozoans' microbiome have shown the presence of abundant and diverse bacterial communities that differ between species and bear little resemblance to the surrounding seawater community [36–40]. Weiland-Bräuer and co-workers [41–43] showed that the composition of the associated bacterial community differs significantly between benthic polyps and the pelagic medusa of *A. aurita*, and between body parts, gastrovascular cavity, and mucus on the umbrella surface.

*A. solida* is widely distributed in coastal areas in the north Adriatic Sea and frequently forms large blooms. Large polyp populations were found on the undersides of mussels growing on harbor piers [44] in the inner part of Koper Bay (northern Adriatic). The adult medusa has a white, transparent bell with up to 28 cm in diameter; it has eight marginal lobes and eight sensory organs [10]. The gonads are horseshoe-shaped. The marginal tentacles are numerous and short; the manubrium is cruciform with four slightly folded oral arms [10].

The aim of this study was to provide comparative information on fine surface structures of polyps and medusae of the bloom-forming invasive scyphozoan *A. solida* (Ulmaridae, Semaeostomeae, Scyphozoa). Building on our previous studies on jellyfish microbiomes [36–38], we sought to describe the presence and localization of microbes on the surfaces of polyps and medusae to better understand the microbiome associations of jellyfish. We investigated the specific surface structures, mucus secretions, and associated microbes using scanning and transmission electron microscopy and histological staining.

## 2. Material and Methods

### 2.1. Animals Collection

*A. solida* medusae were collected from Piran Bay (Northern Adriatic) in May 2015. The medusae were sampled individually with a bucket from a boat, and each specimen was immediately placed in a clean plastic box with surrounding mucus and seawater. In the laboratory, the mucus-coated exumbrella surface was cut into approximately $1 \times 1$ cm pieces with a razor blade and fixed in a solution of glutaraldehyde (final concentration 1%)

and paraformaldehyde (final concentration 0.5%) with sterile seawater (filtered through a 0.22 μm polyethersulfone filter and autoclaved) or formaldehyde (final concentration 2%). All fixed samples were stored at 4 °C until further analysis. In addition, the mucus from the jellyfish surface was taken with a sterile syringe and fixed in a solution of glutaraldehyde and formaldehyde, as described above.

Polyps were collected from a population on the piers in Koper Harbor (Bay of Koper, Northern Adriatic) at a depth of 3.5 m by SCUBA divers in March 2015. Before fixation, polyps were anaesthetized with MgCl and stored at 4 °C until further analysis. For observation with scanning electron microscopy (SEM), anaesthetized polyps were fixed with a combination of aldehydes (1% glutaraldehyde and 0.5% paraformaldehyde, pH 7.2).

### 2.2. Histochemistry of Polyps

For the preparation of histological samples of the polyps of *A. solida,* two protocols were used in order to preserve the mucus as much as possible. In the first one, polyps were fixed with 2% paraformaldehyde (2 h) and then transferred to distilled water (24 h). The samples were embedded in the tissue freezing medium (Jung Leica), frozen (−21 °C and −25 °C), and cut in 10–14 μm slices with a Leica CM 1859 cryomicrotome. Sections were then transferred to poly-L-lysine glass slides and allowed to attach to the slides (protocol [45]). They were stored at −20 °C until staining.

The second protocol included fixation with 4% paraformaldehyde (2 h) and rinsing with distilled water (15 min). After fixation, samples were dehydrated in an ascending series of ethanol and xylene, then infiltrated with paraffin wax at 60 °C (changed after 3 hours, then left overnight). Paraffin-embedded samples were poured into moulds and left to harden. Sections (7 μm) were cut with a Leica RM2265 microtome, transferred to water on microscope slides, and dried on a hot plate. Before staining, slides were deparaffinised and hydrated (xylene and descending series of alcohols).

Alcian Blue, which is known to stain scyphozoan mucins, contains acid mucins and mucopolysaccharides [46,47], and so this stain was used on all sections in combination with Gill's Hematoxylin or Nuclear Fast Red to stain the nuclei. Alcian Blue staining was performed in a 1% Alcian Blue solution in a 3% acetic acid following Cohen et al. [48].

After staining, frozen sections were covered with glycerol–gelatine, and paraffin sections were covered with Pertex. All prepared specimens were observed under a light microscope Opton Axioskop, with a Leica DFC290 HD camera (software Leica LAS).

### 2.3. Sample Preparation for Scanning and Transmission Electron Microscopy

We prepared different samples using several methods for scanning (SEM) and transmission electron microscopy (TEM). For SEM, we prepared specimens of a whole polyp, a polyp cut in half to expose the gastrodermis, and epithelia from the surface of the exumbrella of the medusa. To observe microbes associated on the surface of the polyp or in the mucus, the negative staining method was used for TEM.

For observation with SEM, all fixed samples were first rinsed with Na-cacodylate buffer or Na- phosphate buffer (0.1 M), post-fixed with osmium tetroxide (1%) and rinsed 4 × 5 min with miliQ water. The samples were dehydrated through an ethanol series (50%, 70%, 80%, 90%, 3 min per step and 100% three times per 10 min), and gradually substituted by hexamethyldisilazane (HMDS) (HMDS: 100% ethanol 2:1, HMDS: 100% ethanol 1:2, 3 min per step and HMDS, repeated 3 times). After that, the samples were air-dried at room temperature.

Samples of mucus collected simultaneously with the jellyfish were prepared using two methods. In the first method, fixed mucus samples were applied to the surface of a glass microscope slide and the same chemicals and procedures were used to fix, dehydrate, and stain the sample as described above. In the second method, the mucus samples were fixed with a combination of aldehydes as described above, filtered on a membrane filter with a pore size of 0.2 μm (Ø = 25 mm, Millipore), and washed with sterile sea water (five times, 3 min each). The washed samples were dehydrated in a graded series of ethanol

and dried as described above. All SEM samples were mounted on metal stubs and sputter coated with platinum prior to observation by a field emission scanning electron microscope JSM-7500F (Jeol, Japan).

Polyp mucus was also observed by TEM, and two types of samples were prepared. In the first, the mucus was scraped from the gastrovascular cavity of the polyp and in the second, the polyp was macerated without pedal disc. The samples were transferred to the grids, rinsed with miliQ water, and stained with 1% ($w/v$) water solution of Uranyl Acetate. After drying, specimens were imaged using TEM Philips CM 100 (Philips, Amsterdam, The Netherlands) equipped with a digital camera Gatan Orius 200 SC (Gatan Inc., Pleasanton, CA, USA).

## 3. Results

We examined the presence of bacteria on the surface of the polyps, the excised umbrella pieces of the adult medusae, and mucus samples, collected separately from the medusae and polyps, and analyzed with diverse microscopic techniques. The presence of various polysaccharides was examined in the mucus after the staining of ultrathin longitudinal and transverse tissue sections of the polyp with reagents that detect neutral mucin, glycoproteins, and mucopolysaccharides. The structural organization and differences in surface features between the polyp and the umbrella of the adult medusa were examined of our model species *A. solida*.

### 3.1. Polyp Histochemistry and Surface Morphology

The total body length of polyps, including the basal disc, is about 2–3 mm (Figure 1). Polyps have a cruciform hypostome and in most cases possess 16 tentacles, although the number may vary.

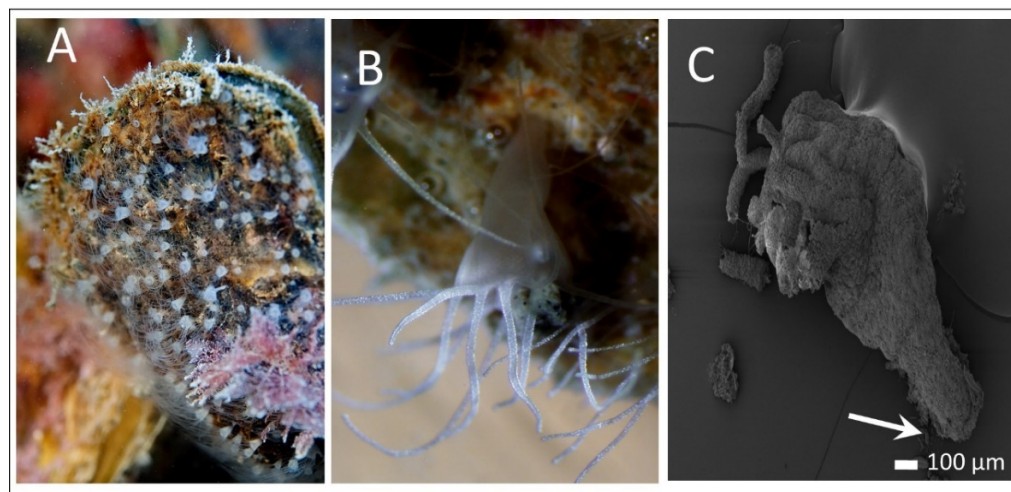

**Figure 1.** The noble pen shell *Pinna nobilis* is covered with the attached polyps of *Aurelia solida*. (**A**) Photograph of polyp (**B**) and scanning electron image (**C**) of the entire polyp showing the body with pedal disc (arrow).

Longitudinal section of a polyp prepared with paraffin sections and stained with Alcian Blue and Gill's Hematoxylin showed the typical cylindrical body with tentacles above the hypostome, the septa in the upper part of the body (Figure 2A,B), and the area of the pedal disc at the base where the polyps attach (Figure 2A).

Higher magnification of the upper part of the polyp showed tubules of numerous discharged nematocysts in the outer and inner parts of the polyp (Figure 2B).

The epithelial structure covers the entire polyp body, from the tentacles to the pedal disc, and forms a boundary layer to the environment. In cross-section of the polyp, two epithelial layers are clearly visible, the epidermis, covering the outer surfaces, and the

thicker gastrodermis lining the gastrovascular cavity. Both layers were separated by very thin mesoglea (Figure 2C).

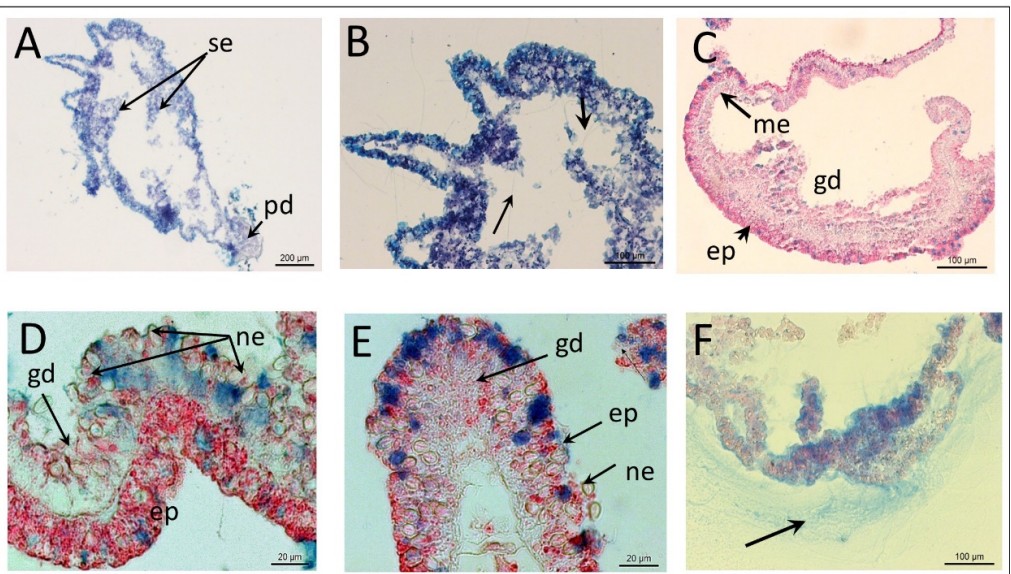

**Figure 2.** Histochemistry of polyps of *Aurelia solida*. (**A**) Longitudinal section of a polyp with tentacles, septa, and pedal disc (arrow); (**B**) Higher magnification of the upper part of the polyp with discharged nematocysts in the outer and inner parts of the polyp (arrow); (**C**) Cross-section of the polyp showing two layers of the body wall and thin intermediate mesoglea; (**D**) Epidermis and gastrodermis contain nuclei (red), mucocytes (blue), and numerous nematocytes on part of the cell wall and (**D**) septa (**E**); (**F**) Deposition of the mucus stained with Alcian Blue (arrow). (**A,B**) Gill's hematoxylin and Alcian Blue (cryosections) (**C–E**) Nuclear Fast Red and Alcian Blue (paraffin sections). ep = epidermis, me = mesoglea, gd = gastrodermis, ne = nematocyst, pd = pedal disc, se = septa.

The epidermis and gastrodermis contain various epithelial cells, glandular cells, and numerous nematocysts on the cell wall and septum (Figure 2D,E). Many nematocysts can be seen, discharged from the surface and inside in the gastrovascular cavity (Figure 2B). Nematocysts are embedded in epidermis and gastrodermis, according to the visible eurytele, with numerous cylindrical forms with rounded ends (Figure 2D,E). Sections, stained with Alcian Blue and Nuclear Fast Red, show the epidermis, with an approximate size between 15–30 µm (Figure 2D), and nuclei stained red and mucocytes stained blue (Figure 2D,E). The mucosecretory material was PAS positive (neutral mucin and glycoproteins) (not shown), and some parts contain dense Alcian Blue–positive reaction products indicative of acidic mucopolysacharides and mucin (Figure 2D,E).

The nematocysts were highly concentrated, especially in the tentacles. The tentacles appeared hollow (Figure 3A) and the multistructural surface of the polyp tentacles can be seen with SEM at high magnification (Figure 3B). The specific spine pattern on the tubule is visible on discharged nematocysts (Figure 3B) and the operculum on non-discharged nematocysts (Figure 3B,C).

The longitudinal section through a polyp revealed various structural features on the surface of the gastrodermis and septum (Figure 4A). The surface of the septum was covered with numerous undischarged nematocysts and cilia (Figure 4B); a dense network of cilia on the gastrodermis is apparent at higher magnification (Figure 4C). In addition to the cilia, high-resolution images of the gastrodermis showed glandular cells, whose openings are clearly visible features on the surface of the septum in the gastrovascular cavity (Figure 4B) and other parts of the gastrodermis (Figure 4D). The surface of the septum was covered with numerous undischarged nematocysts and cilia (Figure 4B), and a dense network of cilia is apparent, especially on the gastrodermis (Figure 4C).

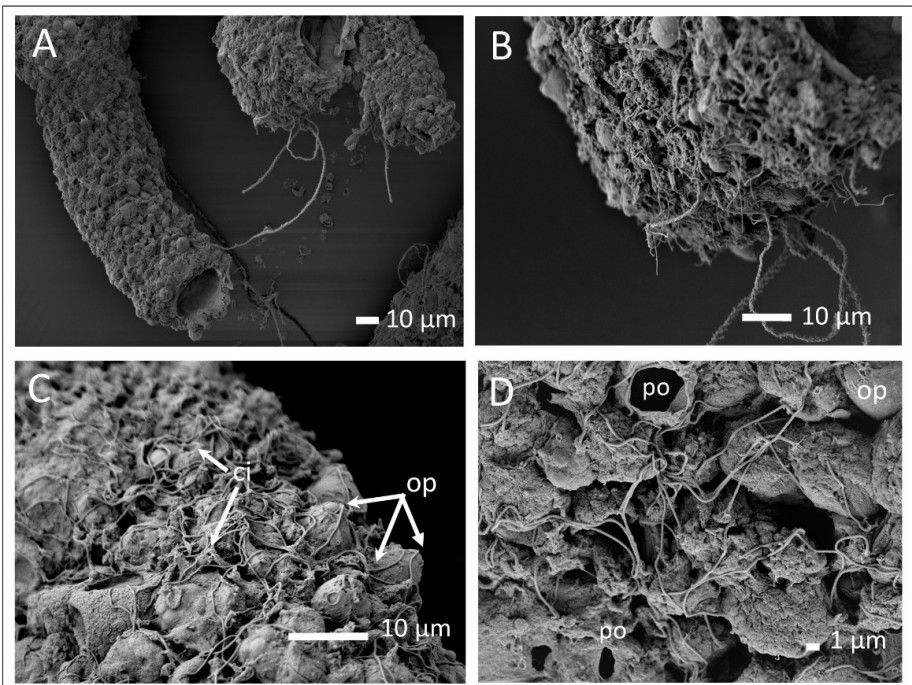

**Figure 3.** Scanning electron micrograph of a polyp of *Aurelia solida*. (**A**) Broken tentacle exposing the interior surface; (**B**) The part of the tentacle with discharged nematocysts showing the specific spine pattern (arrow); (**C**) Part of the surface of the tentacle covered with undischarged nematocysts enclosed by the opercula, various pores, and numerous cilia; (**D**) Section of the surface of the tentacle with pores of glandular cells. ci = cilia, op = operculum, po = pores.

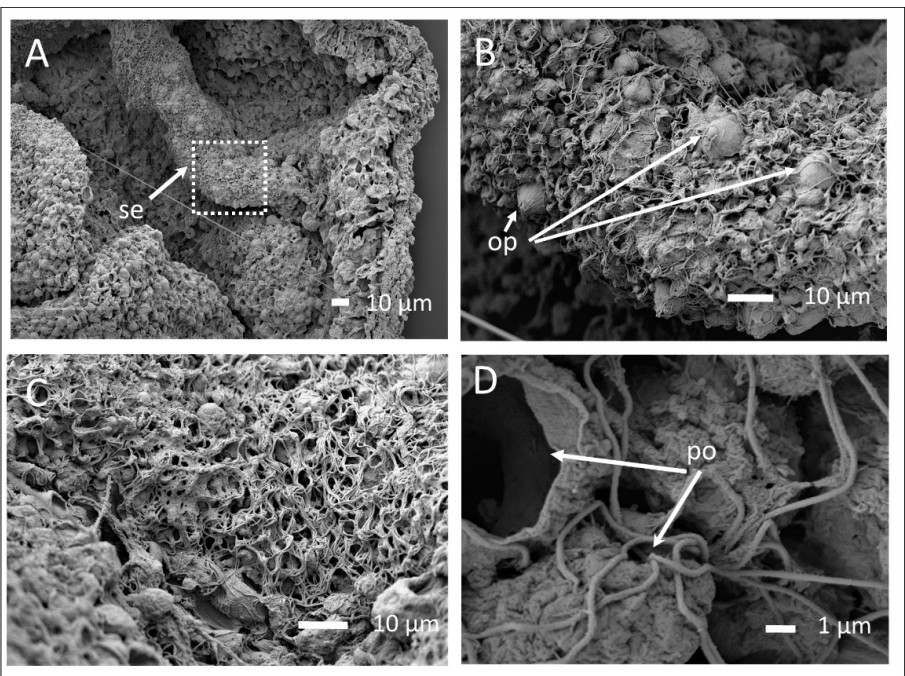

**Figure 4.** High-resolution images of the gastrodermis of *Aurelia solida*. (**A**) Gastrovascular cavity with septum; (**B**) Enlarged view of the surface of the septum (marked white square portion of the septa from image (**A**)) with a large number of undischarged nematocysts and cilia; (**C**) The exposed surface of the gastrodermis is covered with cilia; (**D**) High-resolution image of the gastrodermis showing cilia and pores (arrow) of mucous gland cells. ci = cilia, op = operculum, po = pore, se = septa.

### 3.2. The Exumbrella Surface of the Medusa

Under the light microscope, the exumbrella of the medusa of *A. solida* had a transparent surface with numerous white bulges (Figure 5A,B). Under higher magnification, this even distribution of bulges became visible (Figure 5C). Each bulge consisted of numerous wart-like protrusions about 20–30 μm in diameter (Figure 5D), and numerous crater-like shapes, annular, and circular raised structures with central openings were observed (Figures 4 and 5). Numerous cilia protruding from the epidermis were also observed, and the base of each cilium was surrounded by short microvilli (Figure 5E). Other type of cilia protruding from the crater-like structure was also observed (Figure 5F).

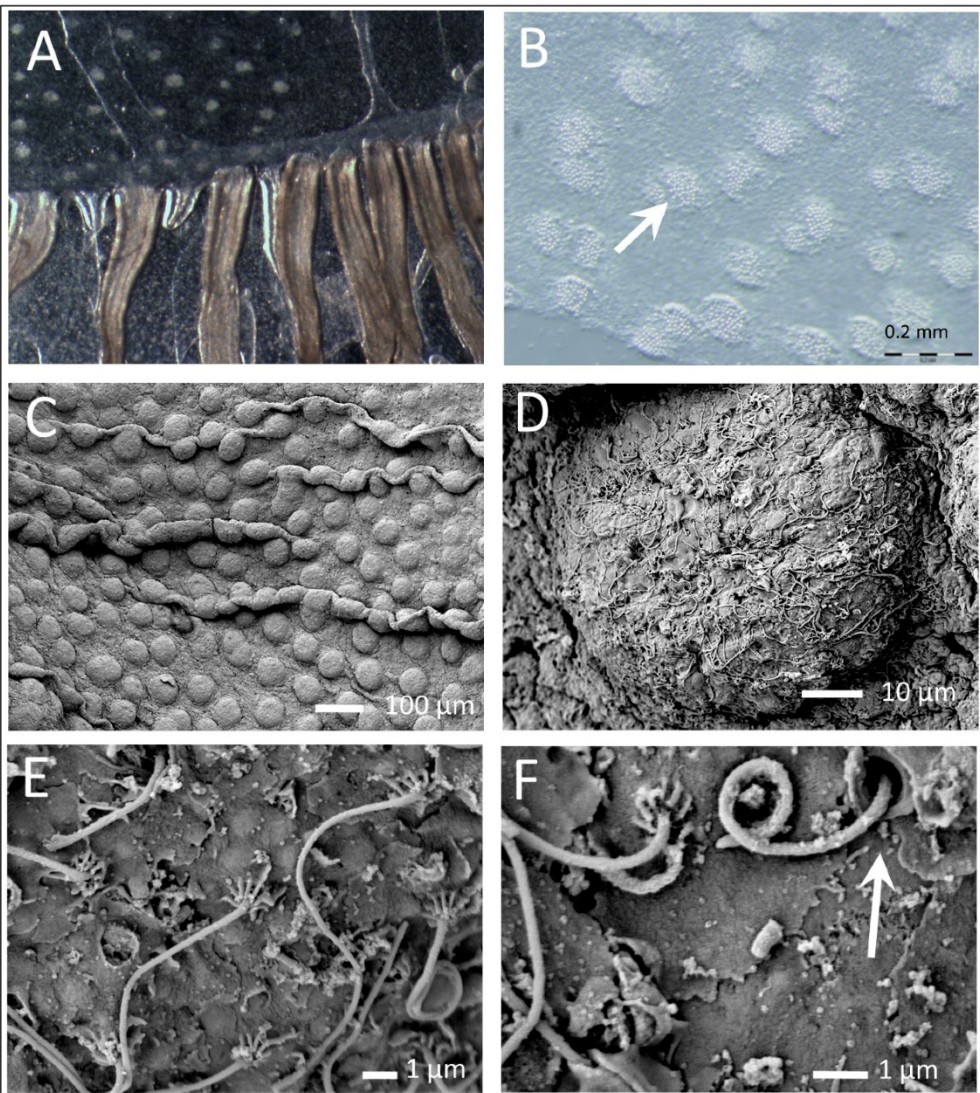

**Figure 5.** Stereomicroscopic and scanning electron micrographs of the surface of part of the exumbrella of an adult medusa of *Aurelia solida*. (**A,B**) White bulges (arrow) on the surface of the medusa bell; (**C**) SEM micrograph of the surface of the medusa exumbrella showing wart-like structures, that comprise each bulge; (**D**) The wart-like structure covered with numerous cilia, crater-like shapes, and circular raised structures with central openings; (**E**) Ciliated tissue surface showing microvilli at the base of cilia (arrow); (**F**) Detail of the two types of cilia, one with the microvilli at the base, the other cilium protruding from the crater-like structure (arrow).

The appearance of the mucus on the exumbrella surface of the medusa *A. solida* was observed using the high-pressure freezing method and no other sample preparation (Cryo

SEM) (Figure 6A) and SEM microscopy (Figure 6B). The mucus appeared on the surface as granular material of nm size surrounding the microvilli of the cilia or concentrated in flocks (Figure 6C,D).

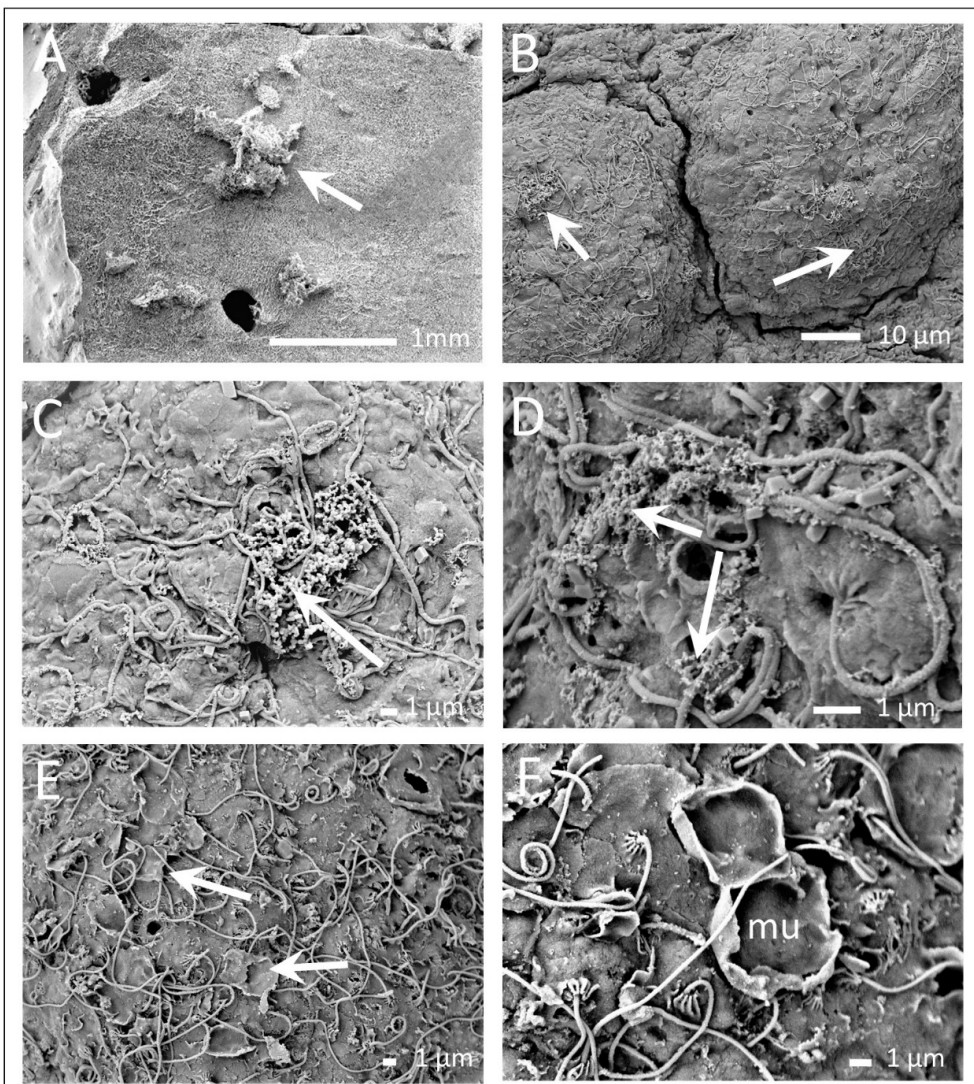

**Figure 6.** SEM micrographs of mucus at the exumbrella surface of the medusa *Aurelia solida*. (**A**) The surface of the medusa with the mucus (arrow) prepared using the high-pressure freezing method and no other sample preparation (Cryo SEM, Philips XL20); (**B**) The surface of the medusa is ciliated and contain granules of mucus (arrow) that are seen clearly at higher magnification (**C**); (**D**) Part of the cluster of granules around cilia and pores; (**E**) The exposed surface of the epidermis reveals multiple crater-like structures and mucus sheets; (**F**) Detail of the ciliated surface with crater-like cowls and sheets of mucus between epithelial cells. mu = mucocytes.

Other interesting structures were crater-like depressions and pores, which we interpret as remains of mucocytes, since the surrounding mucus sheets showed close contact with the epidermis before their detachment (Figure 6F).

### 3.3. Mucus and Associated Microbes

Examining the surface structure of the polyps and the exumbrella surface of the *A. solida*, no attached bacteria were observed. Therefore, the associated bacteria were additionally analyzed from mucus samples collected simultaneously with the jellyfish and mucus scraped from the gastrovascular cavity of the polyp or macerated polyp. SEM

images of the mucus, concentrated on the polycarbonate filters, revealed flocs of mucus material and typical marine rods-shape bacteria of varying cell size (Figure 7A–C). TEM analyses of the scratched mucus and macerated polyp showed different secretion products and structures that were difficult to identify. Bacterial cells of various shapes were seen very clearly, mainly *Vibrio*-like and rod-shaped bacteria (Figure 7D,E) and even bacteriophages (Figure 7F). Based on these results, we hypothesize that the microbial community formed populations on the outer surface of the mucus layer, closely associated with the epidermis.

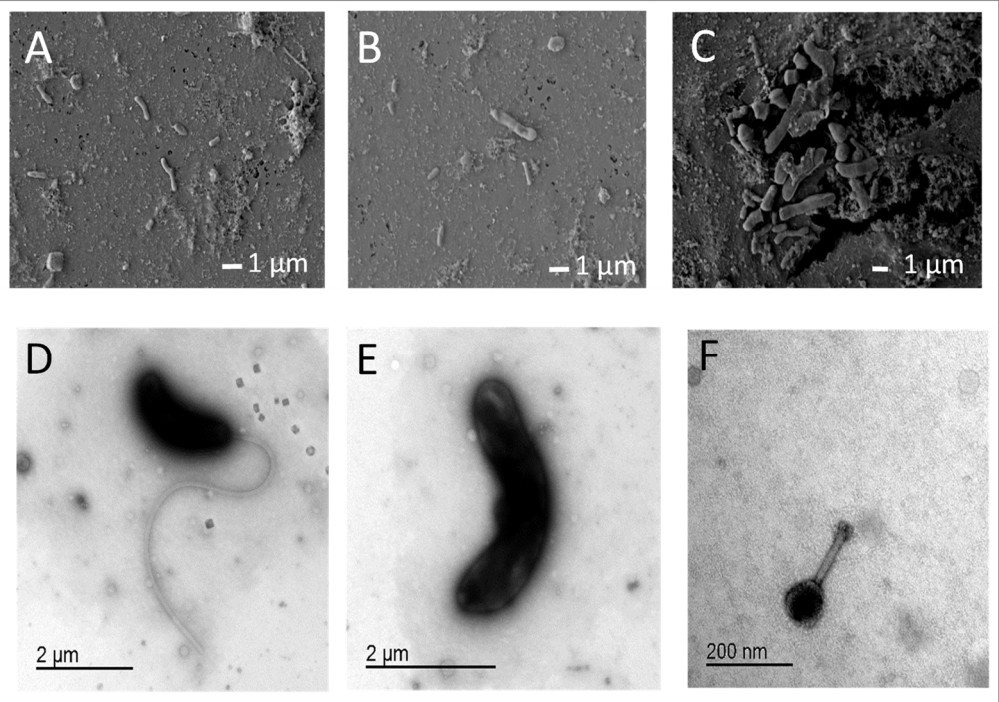

**Figure 7.** Visualization of microbial cells in mucus collected from the surface of *Aurelia solida*. (**A–C**) Bacterial cells in mucus from the surface of adult medusa; (**D–F**) Bacteria in the mucus of scratched and macerated polyp showing different types of bacteria and the bacteriophage.

## 4. Discussion

In the Scyphozoa, polyps and medusae represent two distinct body forms that live in different environments and are characterized by specific macromorphology. Radial tetrameric symmetry, two-layered bodies with outer and inner epithelium and intervening mesoglea, and the presence of stinging apparatus characterize both forms. Their different life strategies, sessile in polyps and planktonic in medusae, condition a different body structure. Differences in body architecture and functioning in specific environments are expected to be reflected in the different surface structure of polyps and medusae of our study organism *A. solida* (Ulmaridae, Semaeostomeae). This species forms large blooms in the Adriatic Sea [9] and is considered non-native to the Adriatic and Mediterranean seas [10].

This study provides evidence for morphologically distinct types of minute surface structures of two life forms (polyp and medusa) of *A. solida*. The importance of structural differences between the epidermal cells of different gelatinous species which allow a transparent body and reduced reflection to be invisible and protected from predators, was shown [4]. The structural difference in the cell composition of the polyp oral tentacles and marginal tentacles of medusa for feeding mechanisms was discussed [49].

As observed in other scyphozoans, the epithelial cells in both polyps and medusae are generally equipped with cilia [16,50], and different cell types such as nematocysts, glandular, muscle, and nerve interstitial cells are generally present [12,15,50–52]. The

anatomy of *A. aurita* has been described, but nothing has been systematically published regarding the ultrastructure of the surface of *Aurelia*. Recently, Chia and coworkers [53] described epithelial cells with a single cilium surrounded by microvilli in tentacles of polyps, muscular layers in epidermis, and vacuolated cells with well-developed Golgi with vesicles.

The epidermis of the polyp and of the gastrodermis were heavily covered with nematocysts, while no nematocysts were observed on the surface of adult medusa. The nematocyst in *Aurelia* has been well studied [54,55], and also for *A. solida* [4] and other jellyfish regularly observed in the north Adriatic Sea [56], as well as the importance of their toxins [57]. Two types of nematocysts, heterotrichous microbasic eurytele and isorhiza haploneme, dominate in polyps of *A. solida* [4]. The presence of isorhizas and the microbasic heteretrotrichous eurytele of the discharged nematocysts were well visible in our SEM images.

The most conspicuous organ of the adult medusa is the exumbrella, and a thick layer of mesoglea, a gelatinous extracellular matrix. The outer surface is covered with thin epidermis composed of flat, usually vacuolated epidermal cells, and can contain mucus cells. The exumbrella surface structures of adult medusa vary among species [4] and among different body parts. The bell rim of *A. aurita* was described in detail [23], the type of cells specialized for the concentration and early digestion of plankton food.

Other specific formations on the exumbrella were large structures without internal content and were widely distributed over the surface of the polyp and medusa of *A. solida*. In the epidermis of scyphopolyps, mucus gland cells of a single type are known, club-shaped, with microvilli, apical cilia, and an apex filled with many polygonal secretory vesicles [12,58]. The mucus is secreted by glandular cells, mucocytes, which are responsible for the constant renewal of the thin layer of mucus covering the surface [50,59]. In our study, the glandular cells were clearly visible in the epidermis and were visualized as openings, pores, by scanning microscopy in both polyps and the medusa. Various stains of cryosectioned samples of polyps also confirm the presence of scyphozoan mucins containing acid mucins and mucopolysaccharides.

Mucus secretion increases under certain conditions, such as physical disturbance, reproduction, nutrition, stress, and also decay [50]. Mucus secretion plays an important role in surface cleaning and serves as a defense mechanism against bacterial overgrowth and predators, in conjunction with toxins released from nematocysts [25,59]. In a recent study, Liu et al. [30] used a combination of proteomic and metabolomics analyses to show that proteins and metabolites (metalloproteinases, serine proteinase inhibitors, toxin-related proteins) are found in the mucus secreted by the jellyfish *A. coerulea*, which have important protective functions. Mucus is an ideal medium to trap and immobilize pathogens and serves as a barrier to microbial infections due to its properties such as elasticity, changeable rheology, and the ability to self-repair [28]. The process of peeling and mucosal restoration might successfully control the abundance of associated bacterial cells, which was observed for corals [60].

The most dominant feature of the surface were the cilia protruding from the epidermis surrounded by microvilli, which covered the entire surface of the polyps and the exumbrella of the medusa. The cilia in the epidermis of the coral produce constant arrays of counter rotating vortices that stir the layer of water up to 2 mm from the epithelial surface, providing enhanced mixing in the diffuse boundary layer with constant motion [61]. The enhanced mixing in this layer has the ability to enhance molecular diffusion transport, nutrient exchange, and oxygen exchange between the coral and the environment [61]. Cilia beating is probably not important for feeding only, but also for cleaning the corals and creating a microhabitat with characteristics that can be considerably different from those of the water column. The importance of mucus secretion and ciliary current in the transfer of food particles to the mouth of adult medusa has been reported for *A. aurita* [23,62].

The interactions between microbes and jellyfish are considered important for the health of the organisms, and visualization of these interactions could lead to a better

understanding of the nature of bacterial association with the host. In our study, the surface of the polyp and medusa of *A. solida*, examined with a SEM, showed no detectable attached microbial cells on the epithelial surface. Rods and *Vibrio*-like bacteria, as well as bacteriophages, were clearly visible in all mucus samples, analyzed separately from the collected mucus samples, from the surfaces of the medusae or the scab samples from the surface and gastrovascular cavity of the polyp. Similar results have been reported for corals [63–66], where microbes did not accumulate on the outer epidermis surface, but attachment was mediated by coral mucus.

The theoretical model of the coral surface layer and microbial interactions was proposed by Ritchie and Smith [67], and Rohwer and Kelly [27]. Ritchie and Smith [67] proposed a stratified mucus layer with metabolites of microbial demineralization of organic exudates. Rohwer and Kelly [27] proposed a model with microbial community colonies and metabolites on the other surface of the mucus layer [26]. Direct observation with high-speed confocal microscopy of live corals and their associated bacterial community also revealed a thick layer on the coral surface where bacteria were concentrated [60].

A review of jellyfish-associated microbial communities and mechanisms of the jellyfish-microbe relationship was presented by Tinta et al. [38] for gelatinous marine plankton belonging to the Scyphozoa, Cubozoa, and Hydrozoa and the phylum Ctenophora. The only publication to date showing microbial attachment to the outer surface of jellyfish was demonstrated by confocal microscopy and fluorescence *in situ* hybridization (FISH) [41]. Weiland-Bräuer and colleagues [41] showed that whole polyps were covered by bacteria that attached or accumulated in the mucus. However, most bacteria were located in the outer surface of the mucus, and only single bacterial cells were detected within the epithelial cells and mesoglea of the polyp (only bacteria of the genus *Mycoplasma*), suggesting a possible endosymbiont interaction. Much more research has been conducted on corals, where microbes have been isolated from the endolytes, digestive tracts, and endosymbiotic zooxanthellae, but most studied coral-association microbes have been obtained from the mucopolysaccharide layer of the coral surface [68].

We were able to detect bacteria only in the mucus samples collected from the surface of the polyp or medusa. Bacteria have different forms and some of them are rather large, compared with the known size of marine bacteria (from 0.2 to 4 μm in size). Recent research suggests that bacteria can change morphology during their life cycle in response to changes in environmental conditions, or simply to increase their ability to colonize distinct hosts [69]. Despite the lack of clarity as to whether the microbes are actually attached to the epithelial layer of the jellyfish or are present in the thin mucus layer, the above evidence indicates that bacterial jellyfish relationships are dynamic and complex and will remain an area of interest in the future.

Numerous results on genetic and metabolic interactions show the fundamental dependence of all biomes on their microbial component and that microbes interact with biotic and abiotic ecosystem components at the molecular/nanometer scale and influence the functioning of complex, large-scale marine ecosystems [70]. Different microbial communities colonize the surface of many marine organisms and form different types of associations [33]. In our study, we examined the structural organization and differences in surface features between the polyp and adult medusa of *A. solida*, as well as the presence of associated bacteria at the surface and in the mucus of the polyp and umbrella. Surface structural differences between polyp and medusa may reflect biomechanical solutions not only for different feeding strategies but also for protection against biofouling/invading microbes, and further work is needed for a more detailed understanding. Since currently, all hypotheses about bacterial associations with jellyfish are rather speculative, we believe that combination of proteomic and metabolomics analyses with state-of-the-art microscopy techniques should be applied to investigate jellyfish microbial interactions, their mechanisms, importance of mucus and its properties of innate immunity, and the relevance of these processes to jellyfish ecology and the marine ecosystem as such.

**Author Contributions:** Conceptualization, V.T. and A.M.; methodology, J.Š., R.K. and M.T.Ž.; analysis, A.F., M.K.K.; writing—original draft preparation, V.T. and A.M.; writing—review & editing V.T., A.M., M.T.Ž., A.F., M.K.K., J.Š. and R.K. All authors have read and agreed to the published version of the manuscript.

**Funding:** The authors acknowledge the financial support from the Slovenian Research Agency (research core funding No. P1-0237).

**Institutional Review Board Statement:** Not applicable.

**Informed Consent Statement:** Not applicable.

**Data Availability Statement:** All data generated and analyzed during this study are included in this article.

**Acknowledgments:** We would like to thanks to our colleagues Tihomir Makovec, and Marko Tadejevič, who provided the polyps of moon jellyfish. We are also grateful to Polona Mrak for her help with preparation and staining of histological samples.

**Conflicts of Interest:** The authors declare no conflict of interest.

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
