# Peer review of "Observations on the Surface Structure of Aurelia solida (Scyphozoa) Polyps and Medusae"

_diversity, doi:10.3390/d13060244_

Round 1

Reviewer 1 Report

Dear editors and authors,

The manuscript (ms) is very interesting by presenting new observations on the surface structure of a scyphozoan species (Aurelia solida). The text is well organized and the ms provides a good background and the objectives are clear. Methods are adequate, and modern as well as traditional techniques have been employed to reach the results. The data in the results section are presented in a concise and clear way which are discussed in accordance with the available literature. The ms is finished with concluding remarks pointing future perspectives for the studies of surface features of scyphozoans and can be expanded to gelatinous zooplankton in general.

I performed some small suggestions directly on the PDF file of the submitted version. These refer mainly to language editing (but note that I am not an English native speaker), text formatting, and literature suggestions.

The figures are essential for the understanding of the results. But I suggest some changes in the captions and size of legends. Those are indicated in the ms PDF file. Perhaps making the images a little bit larger will enhance the details, but it is a suggestion.

References are adequate, including older and brand new ones. But some double checking of formatting is necessary (markings on the ms PDF file).

In my opinion the ms should be accepted after minor corrections and adjustments.

Kind regards...

Author Response

We thank the reviewer for a very helpful and positive review and for the opportunity to submit the manuscript. We tank for comments and concerns.

We address the comments and explain the changes made to the manuscript.

For clarity, we present the reviewer’s comments and our response immediately following them.

We hope that these changes satisfactorily address the concerns and thank you for the opportunity to improve our manuscript.

With best regards,

Review’s comment:

I performed some small suggestions directly on the PDF file of the submitted version. These refer mainly to language editing (but note that I am not an English native speaker), text formatting, and literature suggestions.

Answer:

All suggestions for text formatting and literature were fully considered. The English language has been proofread by a native American proofreader.

Review’s comment:

The figures are essential for the understanding of the results. But I suggest some changes in the captions and size of legends. Those are indicated in the ms PDF file. Perhaps making the images a little bit larger will enhance the details, but it is a suggestion.

Answer:

All proposed changes in the captions and size of legends have been corrected accordingly, and images enlarged to improve detail.

Review’s comment:

References are adequate, including older and brand new ones. But some double checking of formatting is necessary (markings on the ms PDF file).

Answer:

All suggested citations have been included and corrected. Both in the text and in the references, all species names are correctly entered (Italic), but the system keeps changing them.

Reviewer 2 Report

Dear Author,

the manuscript deals with a very interesting topic.

Additional detailed comments are directly reported in the attached Word file.

Best regards.

Author Response

We thank the reviewer for a positive review and for the opportunity to submit the manuscript. We tank for comments and concerns.

We address the comments and all suggestions have been corrected directly  in attached file. Both in the text and in the references, all species names are correctly entered (Italic), but the system keeps changing them.

We hope that these changes satisfactorily address the concerns and thank you for the opportunity to improve our manuscript.

With best regards, Valentina Turk

Reviewer 3 Report

The paper is fine with me, I think it adds significantly to previous papers on the mucus of cnidaria. I have only one concern regarding the vibratile structures of the epithelial cells that the authors call cilia. They are obviously flagella, from the pictures. They are long and there is just one per cell. This is consistent with the definition of flagellum. I advise to read Werner's textbook on Cnidaria Lehrbuch der Speziellen Zoologie, Band I, 2 teil: Cnidaria etc. printed by Gustav Fisher in 1984. In many textbooks planulas and epithelia of cnidarians are labeled as "ciliated" but they are flagellated instead. I am not a histologist, so maybe the terminology has changed and cilia and flagella are considered as synonyms. But I keep finding a distinction between the two in zoology textbooks, even the recentmost. So, my only concern regards the possible confusion between cilia and flagella, which is not a negligible one, in a paper like this. Sometimes taxa that should be in italics (genus and species) are not. Also in the literature, taxa should be written correctly, e.g. Aurelia Solida instead of Aurelia solida. Aurelia aurita appears as Aurelia Aurita several times in the literature.  I let the burden of finding all these taxonomic mistakes to the authors. 

Author Response

We thank the reviewer for a positive review and for the opportunity to submit the manuscript. We address the comments and explain the changes made to the manuscript. For clarity, we present the reviewer’s comments and our response immediately following them.

We hope that these changes satisfactorily address the concerns and thank you for the opportunity to improve our manuscript.

With best regards,

Valentina Turk

Review’s comment:

The paper is fine with me, I think it adds significantly to previous papers on the mucus of cnidaria. I have only one concern regarding the vibratile structures of the epithelial cells that the authors call cilia. They are obviously flagella, from the pictures. They are long and there is just one per cell. This is consistent with the definition of flagellum. I advise to read Werner's textbook on Cnidaria Lehrbuch der Speziellen Zoologie, Band I, 2 teil: Cnidaria etc. printed by Gustav Fisher in 1984. In many textbooks planulas and epithelia of cnidarians are labeled as "ciliated" but they are flagellated instead. I am not a histologist, so maybe the terminology has changed and cilia and flagella are considered as synonyms. But I keep finding a distinction between the two in zoology textbooks, even the recentmost. So, my only concern regards the possible confusion between cilia and flagella, which is not a negligible one, in a paper like this.

Answer:

The terminology of the above structures in cnidarians is very diverse. In functional morphology the term cilia is usually used, and even more frequently quinocilia, to distinguish them from stereocilia, which have a different ultrastructure. The term flagellum is more commonly used for gastroderma cells, but not always. For example, in the textbook Spezielle Zoologie (invertebrates) by Westheide and ed. Rieger a polyp scheme (after Koecke, 1982) where there are cells with cilia on the inside (gastrodermis).The difference between a flagellum and a cilia is difficult to define at the ultrastructural level, it is mainly a difference in function. In the standard scheme of cell structure in zoological textbooks depicting the hydre epithelium, the term cilia is also used.

We would suggest using the term cilia in this case, especially since they are rounded with microvilli and the structure is very similar to other organisms. According to Haji's theory, they are cilia because Turbellaria have ciliated epithelium and Placozoa also have ciliated epithelium. We also include few references where the term cilia is used. Chapman (1970) presents a very similar picture of the polyp surface using the term cilia; this is more of an argument for our decision. However, a detailed and systematic analysis is needed.

Chapman, D.M. Reextension Mechanism of a Scyphistoma’s Tentacle. Can. J. Zool. 1970, 48, 931–943, doi:10.1139/z70-168.

Chia, F.-S.; Amerongen, H.M.; Peteya, D.J. Ultrastructure of the Neuromuscular System of the Polyp of Aurelia aurita L., 1758 (Cnidaria, Scyphozoa). J. Morphol. 1984, 180, 69–79, doi:10.1002/jmor.1051800108.

Heeger, T.; Möller, H. Ultrastructural Observations on Prey Capture and Digestion in the Scyphomedusa Aurelia aurita. Mar. Biol. 1987, 96, 391–400, doi:10.1007/BF00412523.

Hirose, E.; Sakai, D.; Iida, A.; Obayashi, Y.; Nishikawa, J. Exumbrellar Surface of Jellyfish: A Comparative Fine Structure Study with Remarks on Surface Reflectance. Zool Sci 2021, 38, doi:10.2108/zs200111.

Review’s comment:

Sometimes taxa that should be in italics (genus and species) are not. Also in the literature, taxa should be written correctly, e.g. Aurelia Solida instead of Aurelia solida. Aurelia aurita appears as Aurelia Aurita several times in the literature.  I let the burden of finding all these taxonomic mistakes to the authors. 

Answer:

Both in the text and in the references, all species names are correctly entered (Italic), but the system keeps changing them.

Reviewer 4 Report

This study provides interesting and important findings of fine surface structure of the medusae and polyps of Aurelia solida in addition to the mucus secreted by them.  As these have been scarcely investigated before, this study is a significant contribution to understanding the ecology of this species and scyphozoans in general.  I recommend publication after proper modification.

  1. Please check the tense. Since this paper is purely descriptive of the morphology, the present tense should be used for the description. Please examine the whole sentences and change the tense accordingly.

  1. I would emphasize a clear separation of each section: please introduce the background and purpose of your study in the Introduction, describe only materials and methods in Materials and Methods, and describe only results in Results. 

L152-166: This part is not Materials and Methods, but rather Results.  Figures should not be used here, but in Results.

L176-179: The part of medusa bloom is not Results.  Move to Introduction.  Fig. 1A should be deleted.  Figs. 1B and 1C should be used in other part of Results.

  1. Figure legends should be self-explanatory. The species name (Aurelia solida) is missing in the legends of Figs. 2 and 4.

  1. There are so many printing errors throughout the text. These should be checked by authors and printers.

Author Response

We thank the reviewer for a very positive review and for the opportunity to submit the manuscript. We tank for comments and concerns. We address the comments and explain the changes made to the manuscript.

For clarity, we present the reviewer’s comments and our response immediately following them.

We hope that these changes satisfactorily address the concerns and thank you for the opportunity to improve our manuscript.

With best regards,

Valentina Turk

Review’s comment:

Please check the tense. Since this paper is purely descriptive of the morphology, the present tense should be used for the description. Please examine the whole sentences and change the tense accordingly.

Answer:

The suggestions to use the tense has been considered and corrected by a native American proofreader.

Review’s comment:

I would emphasize a clear separation of each section: please introduce the background and purpose of your study in the Introduction, describe only materials and methods in Materials and Methods, and describe only results in Results.

Answer: We apologize for the error, as some paragraphs in the text were entered incorrectly. We have corrected them and taken your suggestions into account.

Review’s comment:

L152-166: This part is not Materials and Methods, but rather Results.  Figures should not be used here, but in Results.

Answer: Thank you fro the comment, the Figure 1 is part of the results.

Review’s comment:

L176-179: The part of medusa bloom is not Results.  Move to Introduction.  Fig. 1A should be deleted.  Figs. 1B and 1C should be used in other part of Results.

Answer: According to suggestion, the figure 1A was deleted and replaced with image representing numerous polyps attached to mussel.

Review’s comment:

Figure legends should be self-explanatory. The species name (Aurelia solida) is missing in the legends of Figs. 2 and 4.

Answer: Thank you for pointing this out. The figure legends have been corrected.

Review’s comment:

There are so many printing errors throughout the text. These should be checked by authors and printers.

Answer: All printing errors throughout the text have been corrected and the English language has been proofread by a native American proofreader.